# Prevalence of malaria and associated factors among under-five children in Sherkole refugee camp, Benishangul-Gumuz region, Ethiopia. A cross-sectional study

Abdulmuneim Ahmed[1]◉, Kebadnew Mulatu◉[1]◉*, Berhanu Elfu◉[2]◉

1 Department of Field Epidemiology, College of Medicine and Health Science, Bahir Dar University, Bahir Dar, Ethiopia, 2 Department of Epidemiology and Biostatics, College of Medicine and Health Science, Bahir Dar University, Bahir Dar, Ethiopia

◉ These authors contributed equally to this work.

* kebadmulatu@gmail.com

**Data Availability Statement:** All relevant data are within the paper and its Supporting Information files.

## Abstract

### Background

Under-five year children are the most vulnerable group affected by malaria, they accounted for 61% of all malaria deaths worldwide. Sherkole refugee camp is stratified under high risk for malaria. Knowledge on malaria prevalence and associated factors among under-five children in Sherkole refugee camp is lacking.

### Methods and materials

Institution-based cross-sectional survey was conducted among under-five children in Sherkole refugee camp from October to November 2019. Total sample size was 356. Stratified random sampling technique was employed to select the study participants. Standardized questionnaire was used to collect data. Care Start[TM] Malaria Rapid diagnostic test which detect histidine-rich protein 2 of P. falciparum and plasmodium lactate dehydrogenase of P. vivax was used to diagnose malaria. Bivariate and multivariable logistic regression analysis was done to identify factors associated with malaria.

### Results

A total of 356 participants were included in this study with response rate of 97.5%. The prevalence malaria was 3.9% (95% CI = 2.0–6.2). Outdoor stay at night (AOR = 3.9, 95% CI = 1.14–13.8), stagnant water near to house (AOR = 4.0, 95% CI = 1.14–14.6), and the number of under-five children per household (AOR = 3.0, 95% CI = 1.03–13.0) were found to increase the odds of getting malaria. Whereas, insecticide treated net (ITN)utilization (AOR = 0.22, 95% CI = 0.06–0.61) and Health information about malaria (AOR = 0.29, 95% CI = 0.06–0.65) reduce the odds of getting malaria.

**Funding:** We declare that we do not have any conflict of interests and this study was funded by Bahir Dar University.

**Competing interests:** there is no competing interest.

## Conclusions and recommendations

Malaria remains the major public health problem in Sherkole Refugee camp. Outdoor stay at night, stagnant water near to house, and number of under-five children per household were the risk factors for malaria. Health information dissemination that focuses on avoiding outdoor stay at night, eliminating stagnant water & using ITN, and considering number of under-five children per household during ITN distribution should be take into account.

## Introduction

Malaria is a common infectious disease caused by protozoan parasites of the genus Plasmodium, that have four species to infect humans namely P. falciparum, P. vivax, P. ovales and P. malarie [1]. Malaria is a complex and deadly disease that puts approximately 3.3 billion (40%) people at risk globally [2]. Under-five year children are the most vulnerable group affected by malaria, they accounted for 61% of all malaria deaths worldwide [3]. In Ethiopia, approximately 52 million people (68%) live in malaria risk areas, primarily at altitudes below 2,000 meters [4]. The prevalence of malaria among under-five children in Benishangul-Gumuz region is 14% [5]. Sherkole refugee camp is located at altitude below 2000 meter and stratified under high risk for malaria [6]. Different intervention activities like distribution of Insecticide Treated Nets (ITN), Indoor Residual Spray (IRS), and Health information dissemination have been done in the study area to prevent and control the burden of malaria, but the disease remain the major public health problem and one of the leading causes of morbidity among under five-year children. Though, several interventions had been implemented but ineffective, because it was not based on the local realities and knowledge of factors that affect malaria distribution. Hence, knowing the current prevalence of malaria and its associated factors in the refugee camp has paramount importance to scale up and design appropriate intervention programs. In this regard, there is scarcity of information since there is no previous study done in the study area. Therefore, this study fills the gap by identifying potential risk factors for malaria infection among under five-year children in sherkole refugee camp.

## Methods

### Study setting and population

Institution-based cross-sectional study was conducted from October to November 2019 in Sherkole refugee camp, Benishangul-Gumuz region, Western Ethiopia. The refugee camp is one of the three refugee camps in Assosa zone, located 705 km from Addis Ababa. It is established in 1997 G.C and currently (October 2019), with a total population of 11017 and 1743 under five children [7]. The main citizens or country of origin in the camp includes South Sudanese, Sudanese, Congolese, and Burundi's. It's altitude is 1450 meters above sea level with average annual temperature ranging from 27 $^{o}$C (during cold months) to 45 $^{o}$C (during dry months) [7]. The source population and study population were all under-five children in Sherkole refugee camp. Children who were not residents of the camp and whose family or caregiver were not agreed to participate in the study were excluded from the study.

### Sample size determination and sampling procedures

The sample size was calculated with the assumption of 25.7% malaria prevalence (for prevalence study) which is taken from study done in Pawe District [8], 95% confidence level, 5%

margin of error, and 10% non-response rate. This brought a total sample size of **365** under five children. The refugee camp is stratified into six zones. An updated Zonal-based sampling frame of the under-five children was obtained from Sherkole refugee camp health center, and number of under-five children was identified from each Zone. Based on the number of under-five children, the sample was proportionally allocated for each zone. Then, under-five children were selected by using simple random sampling from each zone.

## Data collection procedures

Independent variables like socio-demographic characteristics, ITN condition and availability, ITN utilization, IRS, presence of stagnant water, outdoor stay at night, housing condition, and Health information about malaria were included in the questionnaires.

After consent and enrollment, the data was collected by using standardized questionnaire. The questionnaire was pre- tested among 18 under-five children in Tsore refugee camp before the actual data collection. The information was collected from mothers or caregivers of the under-five children by face-to-face interview. ITN utilization was measured by asking the caregivers whether the under-five children slept in ITN during the night preceding the survey or not. ITN condition was assessed by observing the currently used ITN. Caregivers were also asked whether their houses got IRS service in the last six months or not. A finger prick was performed using sterile lancets, and using capillary tube, 5 micro liters of whole blood was drawn from each child included in the sampling regardless of sign and symptoms of malaria. The blood sample was tested immediately using Care Start[TM] Malaria RDT which detect histidine-rich protein 2 (HRP2) of Plasmodium falciparum and plasmodium lactate dehydrogenase (pLDH) of Plasmodium vivax. According to the manufacturer, the sensitivity and specificity of the RDT was 98% and 97.5% respectively.

## Data analysis

All data were entered to EPI Info-7 software and then transferred to SPSS-20 software for analysis. Frequencies, proportions and summary statistics were used to describe the study participants and binary logistic regression was used to examine the association of dependent variable with independent variables. And then, all variables with p-value < 0.2 at bivariate logistic regression were entered to multivariable logistic regression analysis to control the potential confounders. Backward Stepwise (Likelihood Ratio) method was used in multivariable logistic regression model. The Hosmer and Lemeshow test was used to assess the model goodness of fit.

## Ethical approval and consent to participate

The study was ethically approved by Bahir Dar University Institutional Review Board, Benishangul-Gumuz Regional Health Bureau, and Administration for Refugee and Returnee Affairs (ARRA) Assosa sub-office. Verbal assent from the children and a written consent from the care givers or families were taken before data collection. In addition to these, a simple oral explanation of the study was offered to the child before performing study related procedures. Children with positive malaria result were linked to Health center in the camp for appropriate treatments. Confidentiality of study subjects was kept.

## Results

### Socio-demographic characteristics

A total of 356 study participants were included in the study. From the total sample size of 365 children, parents of seven children refused to participate and information collected from two

children was incomplete that made the response rate of 97.5%. Among the study participants, one hundred eighty-four (51.7%) were female; their mean age was 29 (±13.23 SD) months. Among these, 44 (12.4%) were less than one year and majority of them n = 196 (55.1%) were from Sudanese. Of them, 177 (49.7%) respondents had more than 2 under five children. Regarding mother or caregiver related socio-demographic characteristics, majority of them n = 314 (88.2%) were married, 118 (33.1%) were with no formal education by educational status, and 279 (78.4%) has no any monthly income (Table 1).

## ITN coverage and condition

ITN coverage of the study population was 228/356 (64%) with an average of 1.77 per household. From households who have ITN, 44.7%, 38.2%, 12.7%, and 4.4% of them possessed one, two, three, and four ITN respectively. Fifty-six (24.6%) of them reported as their ITN have been tear out/damaged/. Regarding the ITN utilization coverage, 212 (59.6%) of the under-five children slept in ITN during the night preceding the survey.

## Malaria prevalence

The overall prevalence of malaria among under-five children in sherkole refugee camp was 14 (**3.9%) (**(14/356) (95% CI = 2.0–6.2). The proportion of plasmodium species was 85.7%and 14.3% for Plasmodium falciparum and Plasmodium vivax respectively. In terms of gender, the prevalence was 6/172 (3.5%) and 8/184 (4.3%) for male and female respectively.

## Factors associated with malaria infection

**Bivariate and multivariable logistic regression analysis.** In bi-variable logistic regression nine variables (family size, having more than two under -five children per HH, presence of

**Table 1. Socio-demographic characteristics of under-five children in Sherkole refugee camp, BG regional state, Ethiopia, 2019 (N = 356).**

| Socio-demographic variables | Category | N (%) |
|---|---|---|
| Sex | Male | 172 (48.3) |
| | Female | 184 (51.7) |
| Age (Months) | <12 | 44 (12.4) |
| | 12–23 | 73 (20.5) |
| | 24–35 | 80 (22.5) |
| | 36–47 | 106 (29.8) |
| | 48–59 | 53 (14.8) |
| Country of origin | South Sudan | 130 (36.5) |
| | Sudan | 196 (55.1) |
| | Congo | 30 (8.4) |
| Family size | <five | 142 (39.9) |
| | ≥five | 214 (60.1) |
| Number of U5 children | One | 195 (54.8) |
| | ≥two | 161 (45.2) |
| Educational status of care giver | College & above | 9 (2.5) |
| | Secondary school | 35 (9.8) |
| | Primary school | 194 (54.5) |
| | No formal education | 118 (33.1) |
| Income status of care giver (monthly) | No income | 279 (78.4) |
| | <50 USD | 49 (13.7) |
| | >50 USD | 28 (7.9) |

ITN, utilization of ITN, outdoor stay at night, presence of stagnant water near to house, IRS service, presence of hole on the wall of the house, and health information about malaria) were selected and entered in to the backward stepwise multivariable logistic regression model. But only five variables showed significant association in the multivariable model after stepwise selection process.

In multivariable analysis, having more than two under-five children per HH (AOR = 3.0, 95% CI = 1.03–13.0), outdoor stay at night (AOR = 3.9, 95% CI = 1.14–13.8), and presence of stagnant water near to house (AOR = 4.0, 95% CI = 1.14–14.6) infection. Utilization of ITN (AOR = 0.22, 95% CI = 0.06–0.61) and health information about malaria (AOR = 0.29, 95% CI = 0.06–0.65) were found to reduce the odds of malaria (Table 2).

## Discussion

In this study, the overall prevalence of malaria was 3.9% which is lower than the studies done in other countries like Ghana [9], Malawi [10], Uganda [11], and in different parts of Ethiopia such as in Tselemt district north Ethiopia [12], Jima town [13], and in BG region which is 13.9% [5]. This difference might be due to the difference in malaria control and prevention programs implemented in the refugee camp. This also might be due to geographical variation. P. falciparum was the predominant species accounted for 85.7% and the rest was P. vivax (14.3%). This finding is similar with the previous studies done in Uganda [11], and in BG region [5], where the predominant species was Plasmodium falciparum. But, different from the studies done in southern Ethiopia [14], and in Jima [13], where the predominant species was Plasmodium vivax. This might be due to variation in epidemiological distribution of plasmodium species in different parts of the world that might be attributed by geo-spatial variability [3].

**Table 2. Multivariable logistic regression analysis of associated factors for malaria, in Sherkole refugee camp, BG regional state, Western Ethiopia, October 16-November 2, 2019 (N = 356).**

| Variables | Category | Malaria | | COR (95% CI) | AOR (95% CI) | p-value |
|---|---|---|---|---|---|---|
| | | yes | No | | | |
| Family size | ≥Five | 12 | 202 | 4.1 (0.92–16.8) | 2.1 (0.4–10.4) | 0.41 |
| | <Five | 2 | 140 | 1 | 1 | |
| # of U5 children/HH | ≥Two | 10 | 151 | 3.2 (0.97–10.2) | 3.0 (1.03–13.0) | 0.046* |
| | One | 4 | 191 | 1 | 1 | |
| Presence of ITN | Yes | 6 | 222 | 0.40 (0.14–1.19) | 0.41 (0.09–1.9) | 0.25 |
| | No | 8 | 120 | 1 | 1 | |
| Utilization of ITN | Yes | 3 | 209 | 0.18 (0.05–0.63) | 0.22 (0.06–0.61) | 0.024* |
| | No | 11 | 133 | 1 | 1 | |
| Outdoor stay at night | Yes | 10 | 122 | 4.5 (1.3–14.6) | 3.9 (1.14–13.8) | 0.031* |
| | No | 4 | 220 | 1 | 1 | |
| Presence of stagnant water near to house | Yes | 10 | 125 | 4.3 (1.4–14.1) | 4.0 (1.14–14.6) | 0.032* |
| | No | 4 | 217 | 1 | 1 | |
| IRS service | Yes | 12 | 323 | 0.36 (0.07–1.6) | 0.48(0.08–1.8) | 0.41 |
| | No | 2 | 19 | 1 | 1 | |
| Presence of hole on the wall of house | Yes | 6 | 81 | 2.4 (0.82–7.1) | 1.8 (0.6–6.3) | 0.34 |
| | No | 8 | 261 | 1 | 1 | |
| Health information about malaria | Yes | 7 | 289 | 0.19 (0.06–0.54) | 0.29 (0.06–0.65) | 0.008* |
| | No | 7 | 53 | 1 | 1 | |

*P value less than 0.05, AOR: Adjusted Odds Ratio, CI: Confidence interval, "1" in COR and AOR indicate the reference category, U5: Under five, HH: Household.

The odds of malaria infection were higher among children who stayed out-door at night than those who do not stay. This finding is in line with the previous studies done in Zimbabwe [15], Armachiho [16], and Dembia [17]. This could be explained by the fact that they are easily exposed to exophagic-exophilic mosquito bite and gets malaria infection [18].

The odds of malaria infection were higher among children who live in the presence of stagnant water near to their house than those who lives in the absence of stagnant water. This finding is also in line with the previous studies done in Southern Ethiopia [14] and Dembia [17]. This can be explained from the fact that they are more exposed to mosquito bites, because these areas are suitable for breeding of mosquitoes around their homes [19].

The odds of malaria infection were higher among children whose family is with ≥two under-five children than children with a family having one under-five children. This might be related with the number of ITN in the household and mothers' care. As if the number of ITN is not enough, other children can't access ITN and mothers tend to sleep with the youngest child in the household. This finding is in contrast with the previous study done in Mozambique that shows no association between malaria infection and number of under-five children per household [20]. This difference might be due to the difference in study situation (refugee setting Vs community setting) and access to ITN between the two area.

Children who utilize ITN had a reduced risk of malaria infection compared to those who do not utilize ITN. This finding is in contrast with the study done in Uganda where the odds among children who had used an ITN significantly increased by 1.33 times as compared to those who did not use [11]. But it is consistent with the previous study done in Southern Ethiopia [14], Nigeria [21] and East Shewa zone of Oromiya regional state [22]. When they use ITN consistently, the risk of getting mosquito bite might be avoided [19]. Hence, ITN utilizations mut be enforced by government. Children whose mother/caregiver had ever heard health information about malaria had a reduced risk of malaria infection compared to those whose mother/caregiver had never heard health information about malaria. This finding is also supported by the previous studies done in Uganda [11] and Mali [23]. However, it is different from the study done in Debre Elias district where hearing health education message about malaria was not statistically associated with malaria infection [24]. This can be explained by the fact that health education about malaria can increase the awareness about malaria transmission and prevention methods that helps to prevent malaria.

## Limitations of the study

The laboratory investigation for malaria was not carried out by golden standard test (Microscopy) because of unavailability of Electricity. In addition to this, there might be socially desirable bias.

## Conclusion and recommendation

Malaria remains the major public health problem in the study area. Outdoor stay at night, stagnant water near to house, and having more than two under-five children per household were risk factors for malaria. Utilization of ITN and health information about malaria were the protective factor. The local government should give focus about health information dissemination about eliminating of stagnant water, ITN utilization, and the risk of outdoor stay at night.

## Supporting information

**S1 File.**
(SAV)

## Acknowledgments

### Declarations

BGRHB, ARRA Assosa sub office, Sherkole refugee camp administration & Health center staffs, data collectors and supervisors for their valuable contribution in this study. Bahir Dar University and Ethiopian Field Epidemiology Training Program. Finlay we would like to acknowledge data collectors and supervisors.

## Author Contributions

**Conceptualization:** Abdulmuneim Ahmed, Berhanu Elfu.

**Data curation:** Abdulmuneim Ahmed, Kebadnew Mulatu, Berhanu Elfu.

**Formal analysis:** Abdulmuneim Ahmed, Berhanu Elfu.

**Investigation:** Abdulmuneim Ahmed, Kebadnew Mulatu.

**Methodology:** Abdulmuneim Ahmed, Kebadnew Mulatu, Berhanu Elfu.

**Project administration:** Abdulmuneim Ahmed, Kebadnew Mulatu.

**Resources:** Abdulmuneim Ahmed.

**Software:** Abdulmuneim Ahmed.

**Supervision:** Abdulmuneim Ahmed, Kebadnew Mulatu.

**Validation:** Abdulmuneim Ahmed, Berhanu Elfu.

**Visualization:** Abdulmuneim Ahmed, Kebadnew Mulatu, Berhanu Elfu.

**Writing – original draft:** Abdulmuneim Ahmed, Kebadnew Mulatu, Berhanu Elfu.

**Writing – review & editing:** Abdulmuneim Ahmed, Kebadnew Mulatu, Berhanu Elfu.

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
