## [Decision Letter · Decision Letter 0]

19 Oct 2020

PONE-D-20-25756

Prevalence of Malaria and Associated Factors Among Under-five Children in Sherkole Refugee Camp, Assosa Zone, , Western Ethiopia, 2019

PLOS ONE

Dear Dr. Mihretie,

Thank you for submitting your manuscript to PLoS ONE. After careful consideration, we felt that your manuscript requires substantial revision, following which it can possibly be reconsidered, thus governing the decision of a “major revision”. Although your manuscript was of interest to the reviewer, major concerns were related to study design and results.  According to the reviewer, the methods were not described in enough details to allow suitably skilled investigators to fully replicate and evaluate the study. In addition, a significant amount of issues should be clarified and/or adjust otherwise the MS’s results may be compromised. Finally, Ethical issues should be described.  

We look forward to receiving your revised manuscript.

Kind regards,

Luzia Helena Carvalho, Ph.D.

Academic Editor

PLOS ONE

Journal Requirements:

3. Thank you for stating the following above the Acknowledgments Section of your manuscript:

'Funding: This study was funded by Bahir Dar University (BDU)'

'no role except funding'

5. Please amend your list of authors on the manuscript to ensure that each author is linked to an affiliation. Authors’ affiliations should reflect the institution where the work was done (if authors moved subsequently, you can also list the new affiliation stating “current affiliation:….” as necessary).

Reviewers' comments:

Reviewer's Responses to Questions

**Comments to the Author**

1. Is the manuscript technically sound, and do the data support the conclusions?

Reviewer #1: Yes

Reviewer #2: Partly

2. Has the statistical analysis been performed appropriately and rigorously? 

Reviewer #1: Yes

Reviewer #2: I Don't Know

3. Have the authors made all data underlying the findings in their manuscript fully available?

Reviewer #1: Yes

Reviewer #2: No

4. Is the manuscript presented in an intelligible fashion and written in standard English?

Reviewer #1: Yes

Reviewer #2: No

5. Review Comments to the Author

Reviewer #1: Title: Prevalence of Malaria and Associated Factors Among Under-five Children in Sherkole Refugee Camp, Assosa Zone, Benishangul-Gumuz Regional State, Western Ethiopia, 2019.

General overview

The manuscript is relevant to malaria control and elimination program. The authors did a good work in combining the interview with collection of biological specimens to diagnose malaria. But they did not comment if the under-five children had fever or diagnosed malaria are asymptomatic. The introductory to the study needs a review to narrow the malaria prevalence in the broad context of Ethiopia to that of the study area. For reproducibility sake, methods and materials section must be re-written for clarity. Some of the results and what was found in the discussions were not mentioned and the authors can look at this systematically for corrections. Table 2 can be split into bivariate and multivariable tables and more tables are allowed. The tables were not cited in the body of the manuscript. Discussion section should be re-written and avoid repeating the results but systematically explain the results study context in the background of previous study. And why this is novel or new things the study added to the body of science. Conclusion should not include what was not found by the study. Finally, the references must be updated to journal standard.

Abstract

Change the “Introduction” subheading to “Background”

The phrase “fewer than five children” was not clear. Is it referring to under-five children?

The methods and materials section lacks clarity and authors should clearly explain the sampling technique, data collection methods, types of RDT used and analysis. The word count is less than 300, so this can be accommodated.

Conclusion and recommendations

The context of the study is Sherkole not the Country. So, this can be corrected. Recommendations should be linked with responsible person. What is “FP utilization?”

Body of the manuscript

Introduction

Line 1 “Background” change to “Introduction”

Line 9 ITN and IRS were abbreviations and used first time in the manuscript and should be written out in full.

Sherkole refugee camp was classified as high-risk area for malaria infection and some interventions have been done with no effect. The authors need to further justify the need for this study. This statement on line 12 “This might be due to the intervention activities that have been taken was not based on the local realities and knowledge of factors that affect malaria prevalence” need to be further explain and cited.

The title of the manuscript looks at factors associated with malaria. The authors will do better by looking at these factors in the context of the Ethiopia and the study setting in this section of the manuscript.

Methods and materials

Study setting

The first paragraph in this section need a sub-heading title “Study setting and population”

Line 23 “currently (October 2019), with a total population of 11017 and 1743 under five children” where did authors find this population projections? It will be good to cite the source. Also, the settings in the context of malaria needs

Line 33 is it “Pawi” or “Pawe”?

Sample size, the calculation is confusing and what the authors meant by this “This brought a sample of 323 and 365 for prevalence and risk factors study, respectively. Based on these, the study with highest sample size was taken and final sample size was 365 under five children” was not clear. It will be good to be rephrased for clarity. Much more of concern was the selection of the participants. How did authors use stratified sampling with proportional to size (PPS) to select from the six zones? Line 38, “Number of under-five children was identified from each Zone”, how? How did you know which of the children to be sampled?

Data collection procedures

The authors collected data by interviewing the mothers or caregivers and biological specimen from under five children. What is the outcome or dependent variable and which variables are independent variables? What are the contents of the questionnaire? How was it collected? Is it face to face, self-administered? paper-based or electronically?

The authors collected 5 microliters of whole blood, from where and how? It will do the audience better if the author can explain the step by step process, what type of bottle was used to collect the blood? How did author or laboratory scientist process the blood for testing? What kind of RDT kits was used? What was the sensitivity or specificity of the test by the manufacturer? How long was the blood kept before testing or was it tested immediately? How did test differentiate the species of Plasmodium?

Data analysis

Line 51 “Descriptive frequency or descriptive analysis”, I saw that the authors performed quantitative data analysis too. This can be explained in this section.

Results

Line 61 – not good to start a statement with number e.g., “184 (51.7%) were female;” kindly rephrased the entire sentence. Also, “177 (49.7%) were with greater than or equal to two under five children.” What do this mean to the authors?

This title, “Table 1: Socio-demographic characteristics of study participants in Sherkole refugee camps, BG regional state, Western Ethiopia, October 16-November 2, 2019 (N=356).” Can better be specific…..characteristics of under-five children in Sherkole refugee camps, Western Ethiopia, 2019” or Mothers or caregivers of under-five children? But some variables is specific to household. So, the authors need to think if this Table 1 is necessary if the content can be explained in the body of the manuscript than a confused and non-specific table.

The use of illiterate in the table needs to be explain. Does it mean no formal education, not able to read and write or what? Also, in the methods, it will be good the authors explain what they meant by “ration” or generally, “how income status was categorized” for international audience and comparable with previous studies.

Line 71 ITN Coverage and Condition. This was never mentioned in the method and data collection section. How was this data collected? How did the author link the ITN data collected to the title of the manuscript? This was not mentioned in the Introduction and Methods sections. So, it looks abnormal mentioning it at the results section. The author will also mention how ITN utilization was measured.

To report malaria prevalence, the authors need to provide denominators for clarity in this statement, “The proportion of plasmodium species was 12 (85.7%) and 2 (14.3%) for Plasmodium Falciparum and Plasmodium Vivax respectively. In terms of gender, the prevalence was 6 (3.5%) and 8 (4.3%) for male and female respectively.” The statement “Proportion of plasmodium species was not clear. Is the author referring to species count or out of the positive cases, the proportion of children affected with P. falciparum? So, kindly provide the denominator for gender differences.

Bivariate and multivariable logistic regression analysis

The authors did not report the bivariate analysis.

ITN utilization and IRS service was also not reported by the author in the methods, descriptive and bivariate analysis but was mentioned at the multivariable analysis. The author may kindly, make necessary adjustments.

Table 1 and 2 were not mentioned in the manuscript.

Discussions

Authors should avoid repeating the results in this section and it was poorly written. The discussion should address two objectives, namely prevalence of malaria in the refugee camp and factors associated with malaria in this context. The authors will need to review and re-write the section.

This statement, “This might be related with the number of ITN in the household and mothers care. As if the number of ITN is not enough, other children can't access ITN and mothers tend to sleep with the youngest child in the household.”, the study did not look at ITN household coverage to be able to come to this conclusion.

Limitations of the study are numerous, what of information bias? ITN utilization? And many more. The authors can look at this.

Conclusion

This should be in the context of the study. This state “In addition to these, distribution of mosquito repellent would be an important intervention option for those who stay outdoor.”, was not in your result nor in the discussion and not feasible to come into the conclusion.

Declaration

“Missing”

Ethical approval

References

Most of the references need to be updated to the journal standard.

Reviewer #2: Under Question 1, there is need for the Authors to clarify how they arrived at the proportions of the different species of Plasmodium. They indicate in the methods section that they did RDT and later mention as a limitation that they could not do Microscopy because of no electricity in the Refugee settlement. However, when they come to the results, they mention proportions of Plasmodium. It is unclear how they got them because RDT is not used to identify species. The Authors could read more Articles 7,9 and 10 in their references for more on this.

Under 4: There are a number of typos and Grammatical errors, the Authors could address, plus, it would be good for them to use a Good Reference manager.

6. PLOS authors have the option to publish the peer review history of their article (what does this mean?). If published, this will include your full peer review and any attached files.

Reviewer #1: No

Reviewer #2: No

---

## [Author Response · Author response to Decision Letter 0]

5 Jan 2021

Response for reviews 

Response for reviews 

First of all, I would like to thank you for all reviewer for their constructive comments. and I incorporate all comments in the manuscript. The comment give for me is very constrictive and important. so, I try to address all the issue that are given to me, but still now if there is anything that must be amended, I am ready to receive your feedback thank you. But for some issue that needs more justification I wrote here under with.

The authors collected data by interviewing the mothers or caregivers and biological specimen from under five children. What is the outcome or dependent variable and which variables are independent variables? 

The outcome variable is malaria (positive/negative) whereas, Independent variables were socio-demographic characteristics, housing condition /refuge/, ITN condition and availability, ITN utilization, IRS, presence of stagnant water, outdoor stay at night, and Health information about malaria were included in the questionnaires

What are the contents of the questionnaire? How was it collected? Is it face to face, self-administered? paper-based or electronically?

The authors collected 5 microliters of whole blood, from where and how? It will do the audience better if the author can explain the step by step process, what type of bottle was used to collect the blood? How did author or laboratory scientist process the blood for testing? What kind of RDT kits was used? What was the sensitivity or specificity of the test by the manufacturer? How long was the blood kept before testing or was it tested immediately? How did test differentiate the species of Plasmodium?

A finger prick was performed using sterile lancets, and using capillary tube, 5 micro liters of whole blood was drawn from each child. The blood sample was tested immediately using Care StartTM Malaria RDT which detect histidine-rich protein 2 (HRP2) of Plasmodium falciparum and plasmodium lactate dehydrogenase (pLDH) of Plasmodium vivax. According to the manufacturer, the sensitivity and specificity of the RDT was 98% and 97.5% respectively.

77 (49.7%) were with greater than or equal to two under five children.” What do this mean to the authors?nThis mean that when the households had two and above />=2/ under-five children 

To report malaria prevalence, the authors need to provide denominators for clarity in this statement, “The proportion of plasmodium species was 12 (85.7%) and 2 (14.3%) for Plasmodium Falciparum and Plasmodium Vivax respectively.

The overall prevalence of malaria among under-five children in sherkole refugee camp was 14(3.9%) (14/356) (95% CI=2.0-6.2). The proportion of plasmodium species was 12/14 (85.7%) and 2/14 (14.3%) for Plasmodium Falciparum and Plasmodium Vivax respectively. In terms of gender, the prevalence was 6/172 (3.5%) and 8/184 (4.3%) for male and female respectively.

Thank you 

Kebadnew 

Thank you

---

## [Decision Letter · Decision Letter 1]

13 Jan 2021

PONE-D-20-25756R1

Prevalence of Malaria and Associated Factors Among Under-five Children in Sherkole Refugee Camp, Benishangul-Gumuz Region, Ethiopia.A cross-sectional study

PLOS ONE

Dear Dr. Mihretie,

Thank you for submitting your manuscript for review to PLoS ONE. After careful consideration, we feel that your manuscript will likely be suitable for publication if the authors revise it to address critical points raised by the reviewer.  According to reviewer, there are some specific areas where further improvements would be of substantial benefit to the readers.  At this time, we strongly suggest a professional copy editing service.

We look forward to receiving your revised manuscript.

Kind regards,

Luzia Helena Carvalho, Ph.D.

Academic Editor

PLOS ONE

Reviewers' comments:

Reviewer's Responses to Questions

**Comments to the Author**

1. If the authors have adequately addressed your comments raised in a previous round of review and you feel that this manuscript is now acceptable for publication, you may indicate that here to bypass the “Comments to the Author” section, enter your conflict of interest statement in the “Confidential to Editor” section, and submit your "Accept" recommendation.

Reviewer #1: (No Response)

2. Is the manuscript technically sound, and do the data support the conclusions?

Reviewer #1: Partly

3. Has the statistical analysis been performed appropriately and rigorously? 

Reviewer #1: Yes

4. Have the authors made all data underlying the findings in their manuscript fully available?

Reviewer #1: Yes

5. Is the manuscript presented in an intelligible fashion and written in standard English?

Reviewer #1: No

6. Review Comments to the Author

Reviewer #1: Prevalence of Malaria and Associated Factors Among Under-five Children in Sherkole Refugee Camp, Benishangul-Gumuz Region, Ethiopia. A cross-sectional study.

General review

This is a much improve manuscript but needs some corrections most especially, it has a lot of typographical and grammatical errors that needs to be corrected.

Abstract

Background: remove ‘s’ from Under-five years (and at other places in the manuscript)

Methods and Materials: M in Multivariable should be in lowercase

Body of the manuscript

Introduction

Line 60 “Though, intervention has been taken, still now it is not effective” This can be re-phrase to read – “Though, several interventions had been implemented but ineffective because….

Methods and materials

Study setting

Line 87 “in to” should be written as “into six”

Data collection procedures

Line 95 – 97- “Independent variables like socio-demographic characteristics, ITN condition and availability, ITN utilization, IRS, presence of stagnant water, outdoor stay at night, housing condition, and Health 97 information about malaria were included in the questionnaires.” This is not enough. Kindly write how the questions asked was framed and how did you measured the ITN availability and conditions, how did you ascertain ITN utilization and IRS. These are standard indicators and good to explain how you measured them. How did you define ITN or LLIN utilization and IRS service? Is it a Yes or No response? Or how?

Line 98 “pre- tested was done in Tsore refugee camp from 5% of the sample size before the actual data collections time” This can be re-written as “The questionnaire was pre-pretested among 18 under-five children in Tsore refugee camp before the actual data collection.”

Line 100 – what age group of the children, “Is it under-five year children?

Line 102 – signs and symptom of what?

Results

Line 131 “177 (49.7%) were with greater than or equal to two under five children”, please re-write to “177 (49.7%) respondents were living in a household with greater than or equal to two under five children”, for clarity.

Line 132 “care giver” is one word “caregiver”

Line 134 ration is not a means of income so it cannot be income status but source of daily feeding. So kindly correct it in the manuscript.

ITN Coverage and conditions—ITN utilization coverage was not reported here but was mentioned in bivariate and so on. Please, good to mention the coverage at univariate level.

Line 147 “The proportion of plasmodium species was 12/14 (85.7%) 148 and 2/14 (14.3%) for Plasmodium Falciparum and Plasmodium Vivax respectively.” Not clear. Can this be better written?

Also, am worried if RDT can specifically detect P.vivax as the antigen for P vivax are seen in other four species. Only microscopy can do this. This is one of the limitations of the study.

Line 158 “number of under five children” it will be good to have having more than 2 under-five children per household.

Line 161 - Also, “health information about malaria” this was not clear…do you mean awareness or what? ITN utilization should be defined.

Discussion

This section needed a revisit by the authors. There are lots of grammatical errors, misused of alphabets and results repeated.

Line 201 LLIN was first used here and not written in full.

References

Most of the references need to be updated to the journal standard.

7. PLOS authors have the option to publish the peer review history of their article (what does this mean?). If published, this will include your full peer review and any attached files.

Reviewer #1: No

---

## [Author Response · Author response to Decision Letter 1]

25 Jan 2021

Clarification for issues raised on submitted manuscript

First of all, I would like to thank you for all reviewer for their constructive comments. and I incorporate all comments in the manuscript, and for some feathers that needs some sort of explanation, I try to reason out here below. 

1. The issue of RDT 

The type of RDT used to test the blood sample in this study is Care StartTM Malaria RDT.

It detect histidine-rich protein 2 (HRP2) of Plasmodium falciparum and plasmodium lactate dehydrogenase (pLDH) of Plasmodium vivax. So, it can separately detect antigens of P. falciparum and P. vivax. 

According to the manufacturer, the sensitivity and specificity of the RDT was 98% and 97.5% respectively.

2. Regarding Health information about malaria

This means caregivers/mothers who have previous health information/awareness about malaria’s mode of transmission and ways of prevention.

3. Insecticide treated net (ITN) utilization was measured by asking the caregivers whether the under-five children slept in ITN during the night preceding the survey or not. INT condition was assessed by observing the currently used INT. Caregivers were also asked whether their houses got Indoor Residual Spray (IRS) service in the last six months or not

---

## [Decision Letter · Decision Letter 2]

28 Jan 2021

Prevalence of Malaria and Associated Factors Among Under-five Children in Sherkole Refugee Camp, Benishangul-Gumuz Region, Ethiopia.A cross-sectional study

PONE-D-20-25756R2

Dear Dr.  Mihretie,

We’re pleased to inform you that your manuscript has been judged scientifically suitable for publication and will be formally accepted for publication once it meets all outstanding technical requirements.

Kind regards,

Luzia Helena Carvalho, Ph.D.

Academic Editor

PLOS ONE

Additional Editor Comments (optional):

Reviewers' comments:

Reviewer's Responses to Questions

**Comments to the Author**

1. If the authors have adequately addressed your comments raised in a previous round of review and you feel that this manuscript is now acceptable for publication, you may indicate that here to bypass the “Comments to the Author” section, enter your conflict of interest statement in the “Confidential to Editor” section, and submit your "Accept" recommendation.

Reviewer #1: All comments have been addressed

2. Is the manuscript technically sound, and do the data support the conclusions?

Reviewer #1: Yes

3. Has the statistical analysis been performed appropriately and rigorously? 

Reviewer #1: Yes

4. Have the authors made all data underlying the findings in their manuscript fully available?

Reviewer #1: Yes

5. Is the manuscript presented in an intelligible fashion and written in standard English?

Reviewer #1: Yes

6. Review Comments to the Author

Reviewer #1: (No Response)

7. PLOS authors have the option to publish the peer review history of their article (what does this mean?). If published, this will include your full peer review and any attached files.

Reviewer #1: No

---

## [Editor Report · Acceptance letter]

2 Feb 2021

PONE-D-20-25756R2 

Prevalence of Malaria and Associated Factors Among Under-five Children in Sherkole Refugee Camp, Benishangul-Gumuz Region, Ethiopia. A Cross-sectional study 

Dear Dr. mihretie:

I'm pleased to inform you that your manuscript has been deemed suitable for publication in PLOS ONE. Congratulations! Your manuscript is now with our production department. 

Kind regards, 

on behalf of

Dr. Luzia Helena Carvalho 

Academic Editor

PLOS ONE